# Before and after Endovascular Aortic Repair in the Same Patients with Aortic Dissection: A Cohort Study of Four-Dimensional Phase-Contrast Magnetic Resonance Imaging

**DOI:** 10.3390/diagnostics11101912

**Published:** 2021-10-15

**Authors:** Chien-Wei Chen, Yueh-Fu Fang, Yuan-Hsi Tseng, Min Yi Wong, Yu-Hui Lin, Yin-Chen Hsu, Bor-Shyh Lin, Yao-Kuang Huang

**Affiliations:** 1Department of Diagnostic Radiology, Chia Yi Chang Gung Memorial Hospital, Putzu City 61363, Taiwan; chienwei33@gmail.com (C.-W.C.); ymsc10014@gmail.com (Y.-C.H.); 2Department of Diagnostic Radiology, Chang Gung University, Taoyuan 33302, Taiwan; 3Department of Thoracic Medicine, Linkou Chang Gung Memorial Hospital, Taoyuan 33302, Taiwan; dr.fang.yf@gmail.com (Y.-F.F.); 8802003@cgmh.org.tw (Y.-H.T.); 4Department of Thoracic Medicine, Chang Gung University, College of Medicine, Taoyuan 33302, Taiwan; 5Division of Thoracic and Cardiovascular Surgery, Chia Yi Chang Gung Memorial Hospital, Putzu City 61363, Taiwan; mynyy001@gmail.com (M.Y.W.); vw200162@gmail.com (Y.-H.L.); 6Division of Thoracic and Cardiovascular Surgery, Chang Gung University, Taoyuan 33302, Taiwan; 7Institute of Imaging and Biomedical Photonics, National Yang Ming Chiao Tung University, Tainan 71150, Taiwan; borshyhlin@gmail.com; 8Department of Medical Research, Chi-Mei Medical Center, Tainan 30010, Taiwan

**Keywords:** phase contrast, four dimensional, aortic dissection, endovascular repair, malperfusion, magnetic resonance imaging

## Abstract

(1) Background: We used four-dimensional phase-contrast magnetic resonance imaging (4D PC-MRI) to evaluate the impact of an endovascular aortic repair (TEVAR) on aortic dissection. (2) Methods: A total of 10 patients received 4D PC-MRI on a 1.5-T MR both before and after TEVAR. (3) Results: The aortas were repaired with either a GORE TAG Stent (Gore Medical; n = 7) or Zenith Dissection Endovascular Stent (Cook Medical; *n* = 3). TEVAR increased the forward flow volume of the true lumen (TL) (at the abdominal aorta, *p* = 0.047). TEVAR also reduced the regurgitant fraction in the TL at the descending aorta but increased it in the false lumen (FL). After TEVAR, the stroke distance increased in the TL (at descending and abdominal aorta, *p* = 0.018 and 0.015), indicating more effective blood transport per heartbeat. Post-stenting quantitative flow revealed that the reductions in stroke volume, backward flow volume, and absolute stroke volume were greater when covered stents were used than when bare stents were used in the FL of the descending aorta. Bare stents had a higher backward flow volume than covered stents did. (4) Conclusions: TEVAR increased the stroke volume in the TL and increased the regurgitant fraction in the FL in patients with aortic dissection.

## 1. Introduction

Intramural hematoma, perforated aortic ulcers, and type A and type B aortic dissection (AD) have been described as acute aortic syndromes [1,2,3,4]. Patients with medically treated AD remain at significant risk for late adverse events. A recent study recognized that the increased aortic diameter, increased false lumen extent, and forming thrombosis within false lumen were strongly associated with late adverse events [5]. Thoracic endovascular aortic repair (TEVAR) has been used to reduce the growth of the dissecting aortic aneurysms in acute aortic syndrome. However, the effect of the TEVAR impact on hemodynamics is seldom mentioned. Thus, there is a clinical need for a diagnostic tool to assess the risk of false lumen growth to identify patients who may benefit most from prophylactic repair.

Presently, the main imaging modality for detecting aortic diseases is computed tomography angiography (CTA). However, that technique requires the use of contrast media and causes radiation exposure [6,7,8,9]. Compared with conventional angiography and CTA, contrast-enhanced magnetic resonance angiography has higher sensitivity for the characterization of blood vessel pathology. Magnetic resonance imaging (MRI) does not need radiation exposure, but the contrast agents used can still have undesirable effects [10,11]. Further evaluation with contrast-enhanced cross-sectional imaging modalities, such as CT and MRI, is often used to evaluate aortic pathology. However, the major challenge is estimating the proper acquisition time for optimal contrast opacification of the target vessel [12,13]. Four-dimensional phase-contrast MRI (4D PC-MRI) is a non-invasive process that measures blood flow velocity and enables the calculation of the blood flow volume and flow pattern. In addition, 4D PC-MRI can provide detailed visualization of complex blood flow patterns related to healthy and pathological hemodynamics [14]. Thus, it has the potential to quantitatively measure hemodynamics by drawing the region of interest on the two-dimensional PC-MRI image. This analysis method that can quantify the phase-contrast parameters of the region of interest is also called quantitative PC-MRI (QFlow) [15]. Currently, the QFlow technique has been used in research related to cerebrospinal fluid, aorta, and peripheral vascular disease [16,17,18,19,20].

Some evidence has demonstrated that an excess of false lumen inflow relates to increasing pressurization of the false lumen, which promotes the growth of the dissecting aortic aneurysms [21,22,23,24,25,26]. In vitro studies have shown that false lumen pressurization depends to a large extent on the location and cumulative size of the tear [21,22,23,24]. Therefore, it is an essential predictor in the clinical evaluation of chronic aortic dissection. Despite the importance of false lumen pressurization, in vivo techniques to directly measure false lumen pressurization require invasive catheterization, which is rarely performed and potentially hazardous. In vivo studies using image-based measurements reported that flow patterns and flow parameters such as velocity, pressure, and wall shear stress may be potential predictors of aortic dissection [25,26]. However, there is still a great need for clinical application technology to quantify false lumen pressure and hemodynamic abnormalities to facilitate the translation of these experimental results into clinical care.

Our previous clinical study on 4D PC-MRI revealed that potential stent interference and stainless grafts should be avoided [17]. In this study, we used 4D PC-MRI to verify the impact of TEVAR in the same patients with aortic dissection, focusing on hemodynamic changes using the QFlow technique.

## 2. Materials and Methods

### 2.1. Patients

The Institutional Review Board of Chang Gung Memorial Hospital approved this study (number: 201801448B0-1808310074). All patients signed informed consent forms before undergoing examinations. We collected the data of patients who underwent 4D PC-MRI for aortic pathology at Chiayi Chang Gung Memorial Hospital, a tertiary hospital between April 2017 and July 2021, and who had a clinical indication for CTA of aortic dissections. Patients were excluded if they used non-MRI-compatible ferromagnetic devices, were pregnant, exhibited poor compliance, or had an unstable status that prevented them from lying down for MRI. Initially, 51 patients were evaluated. Among them, 10 had received 4D PC-MRI both before and after TEVAR. All patients underwent CTA with intravenous administration of the contrast medium, and 4D PC-MRI was subsequently performed to assess the patients’ aortic pathology.

### 2.2. MRI Methods

We performed imaging on a 1.5-T MRI scanner (Ingenia Rev R5 V30-rev.02; Philips, Amsterdam, the Netherlands) by using an electrocardiogram gating system, with the patient lying in a supine position. Our team performed anatomical scanning of blood vessels around the aortic dissection areas; three planes were scanned separately, and T2 turbo spin echo scanning was carried out with the following parameters: single-shot mode; time repetition (TR), shortest; echo time (TE), shortest; voxel size, 0.6 × 0.84 × 4 mm^3^; the number of signals averaged (NSA), 1; scan duration, 1 min. Balanced turbo field echo scanning was also performed with identical settings, except the voxel size was instead 1.84 × 1.87 × 8 mm^3^. The axial area included the arch to the abdominal bifurcation level, the coronal area comprised the heart and aorta, and the oblique sagittal field included all aorta and parallel aortic arch. The two-dimensional images helped to understand the type and scope of aortic dissection and were the basis for subsequent 4D PC-MRI with the following parameters: three-dimensional turbo field echo (TFE); TR, shortest; TE, shortest; flip angle, 5°; voxel size, 2.25 × 2.25 × 3 mm^3^; phase-contrast velocity, 120 cm/s; scan duration, 6.02 min. Imaging sections had to include the aortic arch and descending aorta. After scanning, the 4D images were used to determine the anatomical space occupied by the artery. Quantitative flow (QFlow) scanning was then performed on a plane perpendicular to the blood flow with the following parameters: scan technique, TFE PC; TR, shortest; TE, shortest; flip angle, 12°; slice thickness, 8 mm; field of view, 248 × 300; phase-contrast velocity, 200 cm/s; scan duration, 13 s while patients held their breath. Those parameters were used and images were captured without using a gadolinium-based contrast agent. We performed QFlow analysis by drawing the region of interest (ROI) on the false lumens and true lumens at the following vascular segments: the aortic root, aortic arch, descending aorta, abdominal aorta at the level of the diaphragm, and abdominal aorta between the level of the celiac trunk and superior mesenteric artery (SMA) (Figure 1). We set the flow direction from the heart to the legs as forwarding/positive flow. On the contrary, the flow direction from the legs to the heart was set as backward/negative flow.

By drawing the ROI completely covering the vascular lumen, the computer could automatically generate analysis results of various variables. These variables include stroke volume (SV), forward flow volume (FFV), backward flow volume (BFV), regurgitant fraction (RF), absolute stroke volume (ASV), mean flux (MF, stroke distance (SD), and mean velocity (MV). All of the eight QFlow variables are shown as follows:Stroke volume, mL;The net volume of blood that passes through the contour of ROI during one cardiac cycle.Forward flow volume, mL;The volume of blood that passes through the contour of ROI in the positive direction (toward head direction) during one cardiac cycle.Backward flow volume, mL;The volume of blood that passes through the contour of ROI in the negative direction (toward foot direction) during one cardiac cycle.Regurgitant fraction, %;The fraction of the minor flow to the main flow that passes through the contour of ROI, automatically defined by the computer.Absolute stroke volume, mL;The absolute value of forwarding flow volume plus the absolute value of backward flow volume.Mean flux, mL/s;Stroke amount x heartbeat/60 (one cardiac cycle).Stroke distance, cm;The net distance that blood proceeds in the vessel during one cardiac cycle.Mean velocity, cm/s.Stroke distance x heartbeat/60 (one cardiac cycle).

### 2.3. Statistical Analysis

Continuous variables (age and QFlow measurements) were analyzed using an unpaired two-tailed Student’s *t* test or one-way analysis of variance test, and discrete variables (sex, substance usage, comorbidities, and intervention history) were compared using a two-tailed Fisher’s exact test. All statistical analyses were conducted using Data Analysis version 8.0 (Stata Corporation, College Station, TX, USA).

## 3. Results

Between April 2017 and July 2021, we enrolled 51 patients (all men; age: 39–56 years) whose aortic pathologies had been evaluated through 4D PC-MRI at a tertiary hospital. Among them, 10 underwent 4D PC-MRI before and after TEVAR. The time between the symptom onset of aortic dissection to the first MRI ranged from 7 days to 10 months. The 10 patients accepted endovascular aortic repair within three days after the first MRI and then arranged a second MRI for postoperative follow-up. The average time between the two MRIs was 215 days (range, 106–298 days). Regarding the patients’ age, sex, comorbidities, aortic disease, TEVAR indication, previous relevant surgeries, stent type, and time between aortic dissection onset and intervention are listed in Table 1. Almost all of the patients were hypertensive; one had Guillain–Barré syndrome, two had polycystic kidney disease, and two had chronic renal insufficiency. Seven patients (Patients 1–7) received TEVAR for chronic dissecting aortic aneurysm with a graft stent (GORE TAG; W.L. Gore &Associates, Inc., Flagstaff, AZ, USA), and the other three (Patients 8–10) received a Zenith Dissection Endovascular Stent (Cook Medical LCC, Bloomington, IN, USA) for malperfusion syndrome after open repair of acute type A aortic dissection. One patient received superior mesentery artery revascularization with a Gore-covered stent, and one received carotid–carotid artery bypass to facilitate coverage of zone 1 in the aortic arch. All patients recovered uneventfully from TEVAR and then underwent postoperative 4D PC-MRI.

Quantitative hemodynamic analysis was performed on all 10 patients before and after TEVAR. Table 2 demonstrates the QFlow measurements of the same 10 participants with aortic dissection before and after TEVAR. Figure 2 illustrates the stroke volume (SV), forward flow volume (FFV), backward flow volume (BFV), and a regurgitant fraction (RF) in the true and false lumens of aortic dissection before and after TEVAR. After TEVAR, the true lumen had higher SV than before TEVAR from the arch to the abdominal aorta. However, the SV of the false lumen decreased after TEVAR, mainly in the descending aorta. The increasing SV of the true lumen is primarily attributable to BFV augmentation in the descending and abdominal aorta. By contrast, FFV increased only in the aortic arch. After TEVAR, RF, which indicates a nonlaminar flow pattern, was higher in the false lumen and lower in the true lumen, mainly in the descending aorta, indicating that the true lumen had predominantly laminar flow after TEVAR. The nonlaminar flow was higher in the false lumen in the aortic arch after TEVAR.

Figure 3 displays the absolute SV, mean flux, SD, and mean velocity in the true and false lumens of aortic dissection before and after TEVAR. The mean flux exhibited a similar trend to that of the SV in both lumens. After TEVAR, the absolute SD increased in the true lumen, whereas the SD was nearly zero in the false lumen. The mean velocity was similar in both lumens after TEVAR. In conclusion, TEVAR increased the forward flow volume of the true lumen (TL). The SV of the false lumen primarily affected the descending aorta. TEVAR decreased the nonlaminar flow in the true lumen in the descending aorta but increased the RF in the false lumen, and the mean flux increased in the true lumen and decreased in the false lumen of the descending aorta. After TEVAR, the SD increased in the true lumen.

Post-stenting quantitative flow analysis was performed to evaluate the impact on bare and covered stents (Table 3, Figure 4 and Figure 5). Covered stents (GORE TAG) caused greater reductions in the SV, backflow volume, and absolute SV than did bare stents in the false lumen of the descending aorta Figure 4A,C and Figure 5A). Notably, bare stents led to higher backward flow than did the covered stents after TEVAR (Figure 4C). The decrease in mean flux and mean velocity in the false lumen was similar between the covered and bare stents (Figure 5B,D). The SD in the abdominal aorta was higher when covered stents were used than when bare stents were used (Figure 5C). These findings are similar to the results of 4D flow visualizations (Appendix A).

## 4. Discussion

In this study, we observed the immediate hemodynamic impact upon the thoracic endovascular aortic repair by 4D phase-contrast MRI through the following parameters estimating true and false lumen of aortic dissection: stroke volume (SV), forward flow volume (FFV), backward flow volume (BFV), and regurgitant fraction (RF). To reduce interindividual variation, we compared the data in the identical patients before and after TEVAR (Figure 2 and Figure 3). The SV was higher in the true lumen of patients with graft stents than in those with aortic dissection without intervention, and the RF, an indicator of nonlaminar flow, was higher in the false lumen than in the true lumen. Thus, TEVAR increased the forward flow volume of the true lumen (TL). The endovascular aortic stent reduced the nonlaminar flow in the true lumen. We also observed the increase in the regurgitant fraction in the false lumen after TEVAR; this result is similar to prior reports [27,28]. The mean flux increased in the true lumen and decreased in the false lumen of the descending aorta. After TEVAR, the SD increased in the true lumen, indicating more effective blood transport per heartbeat.

CT scanners with additional techniques include dual-energy CT and ECG gating manners improved the quality of obtained CTA aortic images [29,30,31,32]. These advances in the CTA dominated the surgical planning for TEVAR but also the post-TEVAR evaluation. However, in patients with impaired renal function or unstable renal flow due to malperfusion syndrome, contrast media may cause acute renal failure [33]. CTA also causes radiation exposure, and substantial accumulation of this radiation can occur, even in young patients [34,35,36,37]. Contrast-enhanced MRI demonstrated blood vessel pathology well with the administration of gadolinium-based contrast agents (GBCA), which shortens blood longitudinal relaxation (T1). This approach provides images with a high signal-to-noise ratio and high spatial resolution by two modes: single-phase and time-resolved MRA [38]. Single-phase MRA captures vascular images at a single point in time. Time-resolved MRA consists of acquiring multiple images of the volume following contrast injection. Blood flow is used as the intrinsic contrast agent, and the signal is based on an inflow effect. The vessels can be observed most clearly when they are orthogonal to the two-dimensional plane because in-plane vessels sometimes experience signal loss [36,37].

The new technique of 4D PC-MRI can, in a single scan, acquire flow information of the entire aortic volume over time [39]. In 4D PC-MRI or 4D flow MRI, the phase contrast, which encodes flow information in all three spatial directions within a large volumetric field of view, is acquired. Many hemodynamic parameters can be derived from these 4D flow data sets, including wall shear stress, pulse wave velocity, blood flow patterns with streamlines, and pressure differences. Pioneering laboratory research has demonstrated that 3.0-T 4D PC-MRI can be used to evaluate aortic dissection, with a focus on aneurysmal change [40,41]. The 4D PC MRI was then compared with the conventional CTA, with similar interexamination, interobserver, and intraobserver variability of these segmentations [42,43]. Recent 4D PC MRI studies have focused on false lumen pressure and the predicted growth in chronic type B aortic dissection [44,45]. They proposed false lumen flow fraction and maximum systolic flow deceleration rate inking to growth for dissection aortic aneurysm [46]. Researchers who conducted those studies did not identify significant limitations in reproducibility or repeatability that may affect measurements derived from 4D flow manners, which is consistent with our previous experience. We first applied 4D PC-MRI in a clinical setting; thus, 4D PC-MRI could provide similar information to that provided by CTA after open surgery for type A aortic dissections [17,46]. Moreover, 4D PC-MRI is also a reasonable imaging option for young patients and patients with poor renal function. However, the choice of stent affects further 4D PC-MRI evaluation. Imaging artifacts with 4D PC-MRI were minimal when nitinol-based endografts were used (GORE TAG and Cook Zenith Dissection Stents) [17]. Stainless steel endoprostheses should not be chosen if 4D PC-MRI is used as a follow-up modality; no such stent graft was used in the current study.

This study has some technical issues to be discussed. First, it revealed that stroke distance is more effective than stroke volume to reflect the hemodynamic difference after TEVAR. We hypothesize that this is because that stroke volume is more affected by the size of the vascular lumen. According to the algorithm, stroke distance is the net distance blood proceeds in the vessel during one cardiac cycle. Stroke volume is the net volume of blood that passes through the contour of ROI during one cardiac cycle. We observed that the vascular lumens (including both false and true lumens) were variable at a different vascular segment. This variability of luminal size at different vascular segments may affect the predictive power of stroke volume. Second, QFlow analysis revealed that regurgitation fractions in the true lumens are consistently small. However, the backward flow volume is large, and the forward flow volume is small in the true lumens of the descending and abdominal aorta. The regurgitant fraction was automatically calculated as the fraction of the minor flow (usually the flow toward the heart) to the main flow (usually the flow away from the heart) that passes through the contour of ROI of the two-dimensional QFlow scanning. The backward flow (negative direction, toward foot) is the main flow characteristic of true lumens at the descending and abdominal aorta. Thus, the regurgitation fractions are still small. Third, the stroke distance and mean velocity can be negative because that stroke distance and mean velocity reflect the “distance” (the flow direction to the head was set as positive flow) that blood proceeds in the vessels. On the contrary, absolute stroke volume and mean flux are positive because the absolute stroke volume was the absolute value of forwarding flow volume plus the absolute value of backward flow volume, and mean flux reflects the stroke amount.

We used bare stents only in patients with malperfusion syndrome after open repair of type A aortic dissection without a proximal covered stent on the secured proximal landing zone. The SD and backflow volume, although still being observed, were lower when bare stents were used than when covered stents were used (Figure 4C and Figure 5 C). Future studies should assess these hemodynamic parameters to explore their application in clinical practice, including prognostic prediction.

The cost of 4D PC MRI may be a concern in merging this diagnostic tool into daily clinical practice. No contrast medium is required for 4D PC-MRI; thus, it would cost little for our national health care system (<USD250 per examination). With the maturation of the radiologic team, this approach is less time consuming (processing time: 30 min), which enables its application for clinical practice.

Our MRI protocol performed QFlow scanning (perpendicular to blood flow and aortic curve) to obtain two-dimensional images, which contained phase-shifting information. By drawing ROI on the vascular lumens (completely covering the true lumen and false lumen), it can obtain hemodynamic variables for statistical analysis. We set the flow direction to the head as positive flow. On the contrary, the flow direction to the foot was set as negative flow. Thus, our result revealed that the net blood volume (stroke volume, SV) in the aortic root and aortic arch was mainly contributed by the forward flow volume (FFV; toward the head direction). On the other hand, the net blood volume (stroke volume) in descending aorta and abdominal aorta was mainly contributed by the backward flow volume (BFV; toward the foot direction) (Figure 2A–C and Figure 4A–C). This result is reasonable according to this study design and MRI protocol.

### Study Limitations

In this study, we verified the clinical value of applying 4D PC-MRI to characterize aortic pathology. However, this study had some limitations. First, The QFlow measurements presented a large standard deviation, and most of the *p*-values are larger than 0.05, indicating no significant difference between groups. Second, this was a nonrandomized study with only a few patients. Further larger-scale randomized studies should be conducted. Third, although quantitative analysis can yield useful information for determining the optimal therapeutic strategy for complex aortic diseases, further studies on quantitative analysis and streamline computation are required, especially to evaluate the endoleak model and explore its other clinical applications.

## 5. Conclusions

As an approach that does not require the use of radiation or contrast media, 4D PC-MRI is a promising alternative modality for imaging aortic dissection. Moreover, this approach may be especially useful for aortic dissection diagnosis and treatment, especially in patients with malperfusion syndrome of visceral vessels, young patients, and patients with impaired renal function. TEVAR increased the SV in the true lumen and increased the RF in the false lumen in the patients enrolled in this study. Whether bare or covered stents are used can influence hemodynamic parameters in 4D PC-MRI.

## Figures and Tables

**Figure 1 diagnostics-11-01912-f001:**
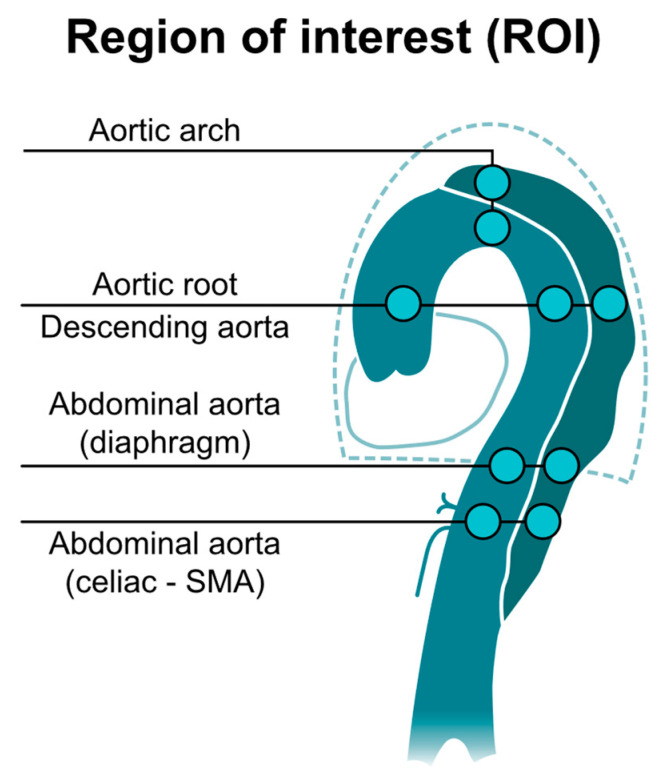
Illustration of QFlow scanning and drawing the region of interest (ROI). The QFlow scanning is performed at four levels to obtain two-dimensional images (perpendicular to blood flow and aortic curve). By drawing ROI on the vascular lumens (completely covering the true lumen and false lumen), eight hemodynamic variables can be obtained for each ROI for the subsequent statistical analysis. The flow direction to the head was set as positive flow.

**Figure 2 diagnostics-11-01912-f002:**
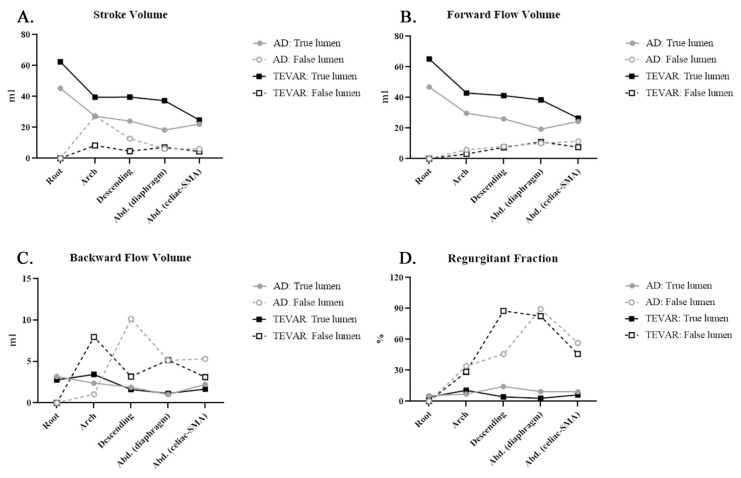
Phase-contrast magnetic resonance imaging (PC-MRI) quantitative flow measurements after thoracic endovascular aortic repair (TEVAR) compared with those before TEVAR (**A**) stroke volume (SV): SV decreased in the false lumen and increased in the true lumen after TEVAR; (**B**) forward flow volume (FFV): FFV increased in the true lumen; (**C**) backward flow volume; (**D**) regurgitant fraction (RF): RF in the aortic arch increased in the false lumen and decreased in the true lumen.

**Figure 3 diagnostics-11-01912-f003:**
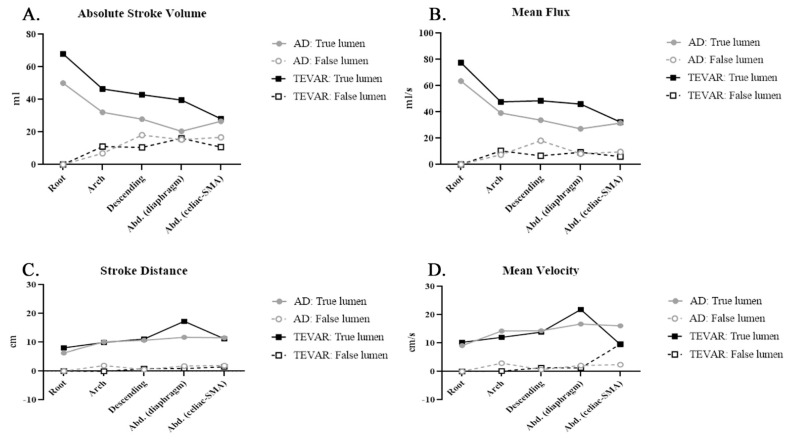
Phase-contrast magnetic resonance imaging (PC-MRI) quantitative flow measurements after thoracic endovascular aortic repair (TEVAR) compared with those before TEVAR: (**A**) absolute stroke volume (ASV): ASV in the true lumen increased in a manner similar to the increase in stroke volume; (**B**) mean flux (MF): MF decreased in the false lumen in the aortic arch; (**C**) stroke distance (SD): SD in the true lumen increased after TEVAR; (**D**) mean velocity (MV): MV in the true lumen increased after TEVAR.

**Figure 4 diagnostics-11-01912-f004:**
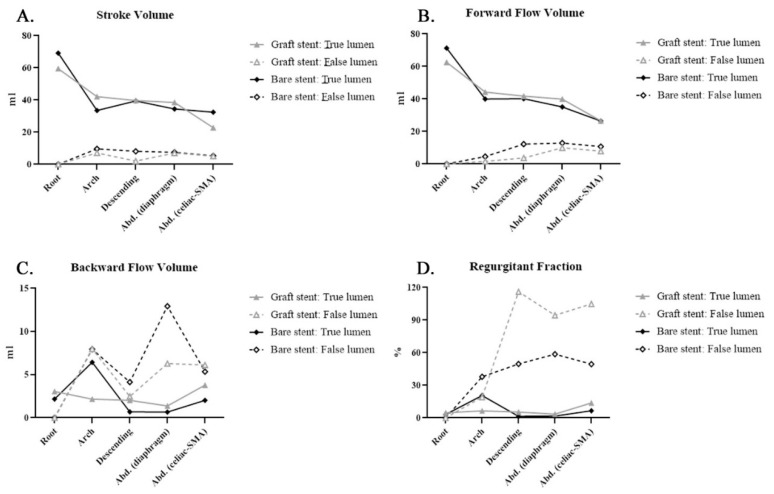
Covered (graft) stent and bare stent: Phase-contrast magnetic resonance imaging (PC-MRI) quantitative flow measurements after thoracic endovascular aortic repair (TEVAR) compared with those before TEVAR: (**A**) stroke volume (SV): SV exhibited a similar distribution between false and true lumens; (**B**) forward flow volume (FFV): FFV exhibited a similar distribution between false and true lumens; (**C**) backward flow volume (BFV): BFV in the false lumen was higher in patients with bare stents than in those with covered stents; (**D**) regurgitant fraction (RF): RF in the false lumen was higher in patients with graft stent than in those with bare stents.

**Figure 5 diagnostics-11-01912-f005:**
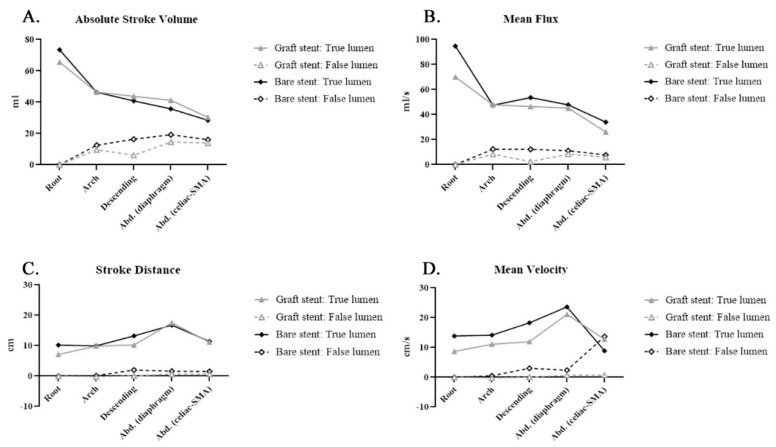
Covered (graft) stent and bare stent: Phase-contrast magnetic resonance imaging (PC-MRI) quantitative flow measurements after thoracic endovascular aortic repair (TEVAR) compared with those before TEVAR: (**A**) absolute stroke volume (ASV): ASV in the false lumen was higher in the bare stent group, indicating fewer communicator occlusions by the bare stent in the thoracic aorta; (**B**) mean flux (MF): MF was higher in the true lumen in patients with bare stents; (**C**) stroke distance (SD): SD in the true lumen was smaller in patients with bare stents than in those with covered stents after thoracic endovascular aortic repair (TEVAR); (**D**) mean velocity (MV): MV was higher in the descending segment but lower in the abdominal aorta in the bare stent group than in the covered stent group after TEVAR.

**Table 1 diagnostics-11-01912-t001:** Demographic of 10 patients receiving 4D PC MRI before and after endovascular aortic repair.

	Age	Sex	Comorbidities	Aortic Disease	Aortic Surgery before This Intervention	Why Intervention	Stent Type	EVAR and Adjuvant Procedure
1	52	M	HTN, PKD	Type B aortic dissection with dilation	No	Aneurysmal change	Gore TAG	No
2	56	M	HTN	Aortic arch dissecting aneurysms	Total arch replacement with branches graft and intraoperative TEVAR.	Aneurysmal change	Gore TAG	No
3	50	M	HTN DM PKD	Type B aortic dissection	Femo-femoral bypass	Aneurysmal change	Gore TAG	Carotid to carotid bypass
4	38	M	HTN renal stone spine surgery	Type B aortic dissection	TEVAR for type B aortic dissection	Severe back pain due to aortic dissection	Gore TAG	No
5	51	M	HTN, CVA	Type B aortic dissection	No	Aneurysmal change of aorta	Gore TAG	No
6	76	M	HTN, GBS	Aortic-dissecting aneurysm	Ascending aortic replacement for acute type A aortic dissection	Aneurysmal change	Gore TAG	Left carotid arterial preservation with chimney procedure by 10 mm Viahbamnn cover stent
7	46	M	HTN, CAD, COPD, CRF	Aortic-dissecting aneurysm	Total arch replacement with branches graft and intraoperative TEVAR.	Severe back pain due to aortic dissection	Gore TAG	No
8	53	M	HTN	Acute Type A aortic dissection	Hemiarch replacement with innominate artery replantation for acute type A aortic dissection	Post-op malperfusion with ischemic bowel	Cook Zenith^®^ dissection endovascular stents	No
9	52	M	HTN	Acute Type A aortic dissection	Ascending aortic replacement for acute type A aortic dissection	Post-op malperfusion with ischemic bowel	Cook Zenith^®^ dissection endovascular stents	SMA by Gore Viahbann 7 mm/5 cm
10	39	M	HTN	Acute Type A aortic dissection	Ascending aortic replacement for acute type A aortic dissection	Post-op malperfusion with ischemic bowel and ileus	Cook Zenith^®^ dissection endovascular stents	No

CAD: coronary arterial disease; CRF: chronic renal failure; CVA: cerebral vascular accident; DM: diabetes mellitus; GBS: Guillain–Barré syndrome; HTN: hypertension; PKD: polycystic kidney disease; SMA: superior mesentery artery; TEVAR: thoracic endovascular aortic repair.

**Table 2 diagnostics-11-01912-t002:** Paired comparison of the QFlow parameters of the same 10 participants with aortic dissection before and after TEVAR.

QFlow	Segment	True Lumen			False Lumen		
		AD	TEVAR	*p*-Value	AD	TEVAR	*p*-Value
SV	Root	45.28 ± 23.89	62.41 ± 22.08	0.122			
	Arch	27.32 ± 12.38	39.51 ± 22.87	0.206	27.32 ± 12.38	8.35 ± 7.50	0.981
	Descending	24.13 ± 13.79	39.64 ± 13.73	0.52	12.76 ± 18.02	4.66 ± 5.27	0.676
	Abdominal (diaphragm)	18.35 ± 15.52	37.3 ± 13.84	0.79	6.16 ± 8.35	7.18 ± 5.98	0.834
	Abdominal (celiac-SMA)	22.07 ± 5.48	24.78 ± 11.41	0.079	6.06 ± 5.00	4.44 ± 3.29	0.072
FFV	Root	46.85 ± 25.96	65.2 ± 22.18	0.1			
	Arch	29.72 ± 13.44	42.96 ± 20.57	0.173	5.86 ± 5.07	3.09 ± 4.31	0.012 *
	Descending	26.02 ± 12.42	41.27 ± 13.38	0.425	7.99 ± 5.60	7.35 ± 6.96	0.052
	Abdominal (diaphragm)	19.38 ± 15.40	38.46 ± 13.89	0.81	10.15 ± 10.40	10.95 ± 7.05	0.504
	Abdominal (celiac-SMA)	24.32 ± 5.39	26.46 ± 12.13	0.047 *	11.37 ± 8.77	7.56 ± 5.10	0.256
BFV	Root	3.2 ± 2.92	2.79 ± 3.82	0.007 *			
	Arch	2.39 ± 2.22	3.45 ± 5.35	0.724	1.05 ± 0.99	7.96 ± 8.17	0.355
	Descending	1.89 ± 2.36	1.62 ± 2.53	0.535	10.14 ± 18.95	3.18 ± 1.49	0.935
	Abdominal (diaphragm)	1.02 ± 1.24	1.16 ± 1.74	0.299	5.14 ± 4.42	5.16 ± 2.76	0.743
	Abdominal (celiac-SMA)	2.25 ± 2.65	1.67 ± 1.16	0.735	5.31 ± 4.19	3.12 ± 2.06	0.717
RF	Root	5.45 ± 4.71	4.2 ± 5.25	0.231			
	Arch	7.01 ± 6.37	10.67 ± 17.85	0.522	34.07 ± 34.80	28.53 ± 33.34	0.718
	Descending	14.4 ± 30.40	4.21 ± 6.45	0.839	45.70 ±40.15	87.64 ± 72.95	0.22
	Abdominal (diaphragm)	9.45 ± 14.41	2.96 ± 4.41	0.883	89.47 ± 59.37	82.52 ± 81.14	0.200
	Abdominal (celiac-SMA)	9.3 ± 10.79	6.25 ± 5.23	0.355	56.58± 24.59	45.75 ± 12.93	0.607
ASV	Root	50.05 ± 25.82	67.99 ± 22.93	0.068			
	Arch	32.11 ± 14.76	46.41 ± 19.50	0.162	6.92 ± 4.70	11.05 ± 7.69	0.811
	Descending	27.92 ± 11.37	42.89 ± 13.49	0.324	18.11 ± 16.52	10.53 ± 8.27	0.946
	Abdominal (diaphragm)	20.4 ± 15.37	39.61 ± 14.14	0.85	15.29 ± 13.09	16.12 ± 7.64	0.175
	Abdominal (celiac-SMA)	26.57 ± 6.49	28.13 ± 12.92	0.182	16.68 ± 12.81	10.68 ± 7.05	0.395
MF	Root	63.53 ± 39.03	77.55 ± 28.39	0.057			
	Arch	39.22 ± 25.08	47.71 ± 26.12	0.147	7.49 ± 8.41	10.34 ± 9.13	0.906
	Descending	33.79 ± 24.18	48.58 ± 15.58	0.32	18.1 ± 26.16	6.59 ± 8.09	0.75
	Abdominal (diaphragm)	27.21 ± 28.77	46.02 ± 15.62	0.797	8.21 ± 10.08	9.28 ± 7.75	0.707
	Abdominal (celiac-SMA)	31.54 ± 9.91	32.31 ± 13.94	0.205	9.62 ± 9.81	6.02 ± 5.06	0.093
SD	Root	6.33 ± 6.62	8.05 ± 2.39	0.05			
	Arch	10.11 ± 3.51	9.91 ± 4.16	0.18	1.95 ± 3.03	−0.06 ± 2.84	0.132
	Descending	10.68 ± 5.46	11.12 ± 3.42	0.018 *	0.59 ± 3.81	0.87 ± 1.52	0.366
	Abdominal (diaphragm)	11.72 ± 6.22	17.29 ± 4.41	0.613	1.78 ± 4.32	0.93 ± 1.32	0.404
	Abdominal (celiac-SMA)	11.54 ± 5.70	11.18 ± 3.98	0.015 *	1.91 ± 1.59	1.48 ± 1.01	0.007 *
MV	Root	9.08 ± 8.92	10.24 ± 3.63	0.033 *			
	Arch	14.26 ± 6.80	12.02 ± 5.65	0.073	2.94 ± 4.48	0.14 ± 3.79	0.098
	Descending	14.37 ± 7.24	13.84 ± 5.04	0.007 *	0.67 ± 5.36	1.3 ± 2.32	0.326
	Abdominal (diaphragm)	16.08 ± 10.83	21.86± 6.30	0.371	2.1 ± 4.61	1.25 ± 1.8.4	0.704
	Abdominal (celiac-SMA)	16.09 ± 8.02	9.49 ± 4.25	0.109	2.4 ± 2.18	9.63 ± 14.5	0.349

AD: aortic dissection (before TEVAR); TEVAR: thoracic endovascular aortic repair; SV: stroke volume; FFV: forward flow volume; BFV: backward flow volume; RF: regurgitant fraction; ASV: absolute stroke volume; MF: mean flux; SD: stroke distance; MV: mean velocity. * *p*-value < 0.05 is defined as statistically significant.

**Table 3 diagnostics-11-01912-t003:** Comparison of the QFlow parameters of the subjects using graft stent (*n* = 7) and bare stent (*n* = 3) after TEVAR.

QFlow	Segment	True Lumen			False Lumen		
		Graft Stent	Bare Stent	*p*-Value	Graft Stent	Bare Stent	*p*-Value
SV	Root	59.51 ± 26.03	69.17 ± 7.9	0.558			
	Arch	42.08 ± 24.54	33.51 ± 21.65	0.617	7.1 ± 6.5	9.61 ± 9.68	0.728
	Descending	39.72 ± 15.83	39.46 ± 9.87	0.98	2.06 ± 1.74	8.13 ± 6.87	0.263
	Abdominal (diaphragm)	38.52 ± 16.54	34.46 ± 4.88	0.696	7.03 ± 6.74	7.5 ± 5.4	0.919
	Abdominal (celiac-SMA)	22.78 ± 7.62	32.4 ± 9.14	0.784	5.13 ± 4.75	5.3 ± 2.25	0.965
FFV	Root	62.56 ± 26.34	71.33 ± 7.23	0.597			
	Arch	44.25 ± 23.49	39. 95± 15.09	0.782	1.62 ± 1.10	4.55 ± 6.25	0.504
	Descending	41.75 ± 15.22	40.14 ± 10.39	0.874	3.67 ± 3.40	12.26 ± 7.84	0.107
	Abdominal (diaphragm)	39.90 ± 16.47	35.11 ± 5.51	0.646	9.95 ± 8.34	12.94 ± 4.00	0.585
	Abdominal (celiac-SMA)	26.56 ± 8.10	26.43 ± 10.35	0.984	7.88 ± 5.65	10.67± 0.68	0.533
BFV	Root	3.05 ± 4.61	2.16 ± 1.04	0.757			
	Arch	2.16 ± 2.53	6.44 ± 9.52	0.271	7.98 ± 8.15	7.94 ± 10.02	0.996
	Descending	2.02 ± 2.91	0.68 ± 1.18	0.457	2.47 ± 1.36	4.13 ± 1.22	0.158
	Abdominal (diaphragm)	1.37 ± 1.98	0.65 ± 1.37	0.581	6.28 ± 2.91	12.93 ± 4.00	0.472
	Abdominal (celiac-SMA)	3.78 ± 3.15	2.02 ± 2.15	0.419	6.12 ± 4.61	5.35 ± 2.93	0.837
RF	Root	4.66 ± 6.29	3.12 ± 1.72	0.697			
	Arch	6.46 ± 8.36	20.51 ± 31.88	0.526	19.17 ± 13.49	37.89 ± 48.32	0.553
	Descending	5.37 ± 7.40	1.49 ±2.57	0.415	116.13 ± 84.54	49.66 ± 38.14	0.268
	Abdominal (diaphragm)	3.5 ± 5.02	1.68 ± 2.91	0.581	94.47± 95.14	58.62 ± 49.22	0.568
	Abdominal (celiac-SMA)	13.78 ± 11.27	6.58 ± 7.27	0.354	104.93 ±75.65	49.46 ± 24.38	0.368
ASV	Root	65.62 ± 27.43	73.51 ± 6.66	0.647			
	Arch	46.41 ± 22.68	46.38 ± 12.95	0.999	9.61 ± 8.82	12.49 ± 7.98	0.697
	Descending	43.77 ± 15.16	40.83 ± 11.02	0.772	6.13 ± 4.91	16.4 ± 8.88	0.104
	Abdominal (diaphragm)	41.25 ± 16.61	35.78 ±6.29	0.604	14.57 ± 9.17	19.22 ± 1.26	0.273
	Abdominal (celiac-SMA)	30.35 ±9.63	28.45 ± 11.85	0.802	13.40 ± 7.46	16.02 ±3.61	0.734
MF	Root	70.16 ± 30.83	94.8 ± 11.79	0.228			
	Arch	47.83 ± 25.74	47.43 ± 32.9	0.984	8.39 ± 7.93	12.28 ± 11.58	0.657
	Descending	46.43 ± 17.35	53.6 ± 11.63	0.537	2.41 ± 2.10	12.18 ± 10.39	0.234
	Abdominal (diaphragm)	45.24 ± 18.75	47.83 ± 6.04	0.826	8.35 ± 8.08	11.14 ± 8.33	0.643
	Abdominal (celiac-SMA)	26.1 ± 8.20	33.98 ± 12.62	0.287	5.94 ± 5.70	7.7 ± 4.60	0.711
SD	Root	7.13 ± 2.02	10.19 ± 1.87	0.056			
	Arch	9.89 ± 3.61	9.94 ± 6.22	0.987	−0.23 ± 0.58	0.1 ± 4.44	0.907
	Descending	10.22 ± 3.75	13.2 ± 1.04	0.226	0.04 ± 0.42	1.98 ± 1.85	0.089
	Abdominal (diaphragm)	17.53 ± 5.17	16.74 ± 2.52	0.811	0.61 ± 1.24	1.58 ± 1.50	0.328
	Abdominal (celiac-SMA)	11.15 ± 2.84	11.42 ± 4.53	0.916	0.61 ± 2.00	1.52 ± 0.90	0.571
MV	Root	8.68 ± 3.11	13.86 ± 1.38	0.027 *			
	Arch	11.1 ± 3.86	14.16 ± 9.43	0.464	−0.22 ± 0.74	0.5 ± 5.91	0.843
	Descending	11.94 ± 4.48	18.28 ± 3.44	0.062	0.03 ± 0.5	3 ± 2.87	0.090
	Abdominal (diaphragm)	21.12 ± 6.74	23.6 ± 6.02	0.599	0.68 ± 1.52	2.39 ± 2.22	0.208
	Abdominal (celiac-SMA)	12.78 ± 3.07	8.89 ± 4.75	0.173	0.71 ± 2.35	13.7 ± 17.88	0.081

TEVAR: thoracic endovascular aortic repair; SV: stroke volume; FFV: forward flow volume; BFV: backward flow volume; RF: regurgitant fraction; ASV: absolute stroke volume; MF: mean flux; SD: stroke distance; MV: mean velocity. Data are presented as the mean ± standard deviation (SD). * *p*-value < 0.05 is defined as statistically significant.

## Data Availability

The data presented in this study are available on request from the corresponding author. The data are not publicly available due to ethical restrictions.

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
