# Peer review of "Before and after Endovascular Aortic Repair in the Same Patients with Aortic Dissection: A Cohort Study of Four-Dimensional Phase-Contrast Magnetic Resonance Imaging"

_diagnostics, 2021, doi:10.3390/diagnostics11101912_

Round 1
Reviewer 1 Report
none
Author Response
[Comment 1]
The design of this research must be improved.
[Answer 1]
Thank for your thoughtful comment. We revised this article with more informative introduction toward study design,

Reviewer 2 Report
This paper describes a cohort of 10 patients that underwent 4D MRI both prior and post TEVAR operation for aortic dissection. 4D flow MRI is a promising technique in enhancing our understanding of aortic dissection hemodynamics. In the current study, flow trends in the true and false lumens were presented and compared before and post TEVAR. Paper concludes that stroke volume, the stroke distance, and forward flow have increased in the true lumen after the TEVAR, while regurgitant fraction has increased in the false lumen. The study is interesting given the fact that it demonstrates flow changes in the same patients before and after the TEVAR, but it will benefit from a more thorough analysis. Below please find some comments that can help further improve the manuscript.
- An introduction is very short currently with some of the claims not referenced (like the evidence for the correlation of false lumen inflow and pressurization). It would benefit from expanding, reworking and adding more references to the claims.
- Please add results of quantitative analysis, including value differences between flow TL pre-post, true vs false lumen, etc and indicate statistical significance. Results of the statistical analysis are not presented even though it is listed in the methods portion.
- Nonlaminar flow in FL detected by 4D MRI post-TEVAR has been previously reported (PMID: 22841438, 22386146), so increased regurgitant fraction is not “paradoxical”.
- Table 1 does not have the time between the onset of AD and MRI that was referenced in the text. Is it the time to the 1st MRI (pre-TEVAR), or time to the post-TEVAR MRI? What was the average time between two MRIs?
- In all graphs, false lumen values at the root location are the same as true lumen values. Shouldn’t they be zero since there is no false lumen at the aortic root?
- Based on Figures 1B & 3B, forward flow in the descending and abdominal aorta segments is close to zero in both true and false lumens. Please comment on how this can happen. Is it the volume per cardiac cycle, or a snapshot at a particular time?
- The paper would benefit from proofreading, especially its Intro part.
- Did the authors detect any endoleak problems in their cohort with post-TEVAR MRI?
Minor:
P1, Line 38-39: incomplete sentence (aortic disease imaging?)
P1, Line 45: … are also a concern (?)
There is no 1.2 part, so it is suggested to omit 1.1 title.
What is meant by back forward flow volume? Backward and forward flow?
Identical patients sound strange.
Author Response
Reviewer 2
[Comment 1]
This paper describes a cohort of 10 patients that underwent 4D MRI both prior and post TEVAR operation for aortic dissection. 4D flow MRI is a promising technique in enhancing our understanding of aortic dissection hemodynamics. In the current study, flow trends in the true and false lumens were presented and compared before and post TEVAR. Paper concludes that stroke volume, the stroke distance, and forward flow have increased in the true lumen after the TEVAR, while regurgitant fraction has increased in the false lumen. The study is interesting given the fact that it demonstrates flow changes in the same patients before and after the TEVAR, but it will benefit from a more thorough analysis. Below please find some comments that can help further improve the manuscript.
An introduction is very short currently with some of the claims not referenced (like the evidence for the correlation of false lumen inflow and pressurization). It would benefit from expanding, reworking and adding more references to the claims.
[Answer 1]
We had expanded the introduction and added references.
[Change]
“Patients with medically treated aortic dissection remain at significant risk for late ad-verse events. A recent study recognized that the increased aortic diameter, increased false lumen extent, and forming thrombosis within false lumen were strongly associated with late adverse events. Thoracic endovascular aortic repair (TEVAR) has been used to reduce the growth of the dissecting aortic aneurysms in acute aortic syn-drome. However, the effect of the TEVAR impact on hemodynamics is seldom be mentioned. Thus, there is a clinical need for a diagnostic tool to assess the risk of false lu-men growth to identify patients who may benefit most from prophylactic repair.”
“4D PC-MRI can provide detailed visualization of complex blood flow patterns related to healthy and pathological hemodynamics. Thus, it given the potential to quantitatively measure hemodynamics. By drawing the region of interest on the two-dimensional PC-MRI image. This analysis method that can quantify the phase-contrast parameters of the region of interest is also called quantitative PC-MRI (QFlow). Currently, QFlow technique has been used in research related to cerebrospinal fluid, aorta, and peripheral vascular disease.
“In vitro studies have shown that false lumen pressurization depends to a large extent on the location and cumulative size of the tear. Therefore, it is an essential predictor in the clinical evaluation of chronic aortic dissection. Despite the importance of false lumen pressurization, in vivo techniques to directly measure false lumen pressurization require invasive catheterization, which is rarely performed and potentially hazardous. In vivo studies using image-based measurements reported that flow patterns and flow parameters such as velocity, pressure, and wall shear stress may be potential predictors of aortic dissection. However, there is still a great need for clinical application technology to quantify false lumen pressure and hemodynamic abnormalities to facilitate the translation of these experimental results into clinical care.” [Introduction]
[Comment 2]
Please add results of quantitative analysis, including value differences between flow TL pre-post, true vs false lumen, etc and indicate statistical significance. Results of the statistical analysis are not presented even though it is listed in the methods portion.
[Answer 2]
We had added Table 2 and Table 3 in this revised manuscript to demonstrate the results of quantitative analysis.
[Comment 3]
Nonlaminar flow in FL detected by 4D MRI post-TEVAR has been previously reported (PMID: 22841438, 22386146), so increased regurgitant fraction is not “paradoxical”.
[Answer 3]
Thanks for your correction. We had deleted the word” paradoxical” and added references to support this result.
[Change]
“The endovascular aortic stent reduced the nonlaminar flow in the true lumen. We also noticed the increase of regurgitant fraction in the false lumen after TEVAR; this result is similar to prior reports.( PMID: 22841438, 22386146)” [Discussion]
[Comment 4]
Table 1 does not have the time between the onset of AD and MRI that was referenced in the text. Is it the time to the 1st MRI (pre-TEVAR), or time to the post-TEVAR MRI? What was the average time between two MRIs?
[Answer 4]
The time between the symptom onset of aortic dissection to the 1st MRI ranged from 7 days to 10 months. The ten patients accepted endovascular aortic repair within three days after the 1st MRI and then arranged the 2nd MRI for post-operative follow-up. The average time between the two MRIs was 215 days (range, 106-298 days). We had added this description in the Result section.
[Change]
“The time between the symptom onset of aortic dissection to the 1st MRI ranged from 7 days to 10 months. The ten patients accepted endovascular aortic repair within three days after the 1st MRI and then arranged the 2nd MRI for post-operative follow-up. The average time between the two MRIs was 215 days (range, 106-298 days).” [Results]
[Comment 5]
In all graphs, false lumen values at the root location are the same as true lumen values. Shouldn’t they be zero since there is no false lumen at the aortic root
[Answer 5]
Thanks for your correction. The false lumen values at the root location should be zero because there is no false lumen at the aortic root among these 10 cases. We had redrawn all the graphs (Figure 1-5) to correct this error.
[Comment 6]
Based on Figures 1B & 3B, forward flow in the descending and abdominal aorta segments is close to zero in both true and false lumens. Please comment on how this can happen. Is it the volume per cardiac cycle, or a snapshot at a particular time?
[Answer 6]
The QFlow scanning is performed at four levels to obtain two-dimensional images (perpendicular to blood flow and aortic curve). By drawing ROI on the vascular lumens (completely covering the true lumen and false lumen), it can obtain eight hemodynamic variables for each ROI for the following statistical analysis. We set the flow direction to the head as positive flow. On the contrary, the flow direction to the foot was set as negative flow.
Stroke volume is the net volume of blood that passes through the contour of ROI during one cardiac cycle. Forward flow volume is the volume of blood that passes through the contour of ROI in the positive direction (toward the head direction) during one cardiac cycle. Backward flow volume is the volume of blood that passes through the contour of ROI in the negative direction (toward the foot direction) during one cardiac cycle.
Thus, the net blood volume (stroke volume, SV) in the aortic root and aortic arch was mainly contributed by the forward flow volume (FFV; toward the head direction). On the other hand, the net blood volume (stroke volume) in descending aorta and abdominal aorta was mainly contributed by the backward flow volume (BFV; toward the foot direction). This result is reasonable according to this study design and MRI protocol. We added Figure 1 and associated descript to clearly explain our study and MRI techniques.
All of the eight Q-Flow variables are shown in Figure, as follows:
(1) Stroke volume, ml. The net volume of blood that passes through the contour of ROI during one R-R-interval.
(2) Forward flow volume, ml. The volume of blood that passes through the contour of ROI in the positive direction (toward head direction) during 1 R-R-interval.
(3) Backward flow volume, ml. The volume of blood that passes through the contour of ROI in the negative direction (toward foot direction) during 1 R-R-interval.
(4) Regurgitant fraction, %. The fraction of the minor flow (usually backward flow) to the major flow (usually forwarding flow) that passes through the contour of ROI, automatic defined by the computer.
(5) Absolute stroke volume, ml. The absolute value of forwarding flow volume plus the absolute value of backward flow volume.
(6) Mean flux, ml/s. Stroke amount x heartbeat / 60 (1 R-R interval).
(7) Stroke distance, cm. The net distance that blood proceeds in the vessel during 1 R-R-interval.
(8) Mean velocity, cm/s. Stroke distance x heartbeat / 60 (1 R-R interval).
“We performed QFlow analysis by drawing the region of interest (ROI) on the false lu-mens and true lumens at the following vascular segments: the aortic root, aortic arch, descending aorta, abdominal aorta at the level of the diaphragm, and abdominal aorta between the level of the celiac trunk and the superior mesenteric artery (SMA) (Figure 1). We set the flow direction to the head as positive flow. On the contrary, the flow di-rection to the foot was set as negative flow.” [Materials and Methods]
“Figure1. Illustration of QFlow scanning and drawing the region of interest (ROI). The QFlow scanning is performed at four levels to obtain two-dimensional images (perpendicular to blood flow and aortic curve). By drawing ROI on the vascular lumens (completely covering the true lumen and false lumen), it can obtain eight hemodynamic variables for each ROI for the following statistical analysis. We set the flow direction to the head as positive flow” [Figure 1]
“By drawing the ROI complete covering the vascular lumen, the computer could automatically generate analysis results of various variables. These variables include Stroke volume (SV), forward flow volume (FFV), backward flow volume (BFV), re-gur-gitant fraction(RF), absolute stroke volume (ASV), mean flux(MF, stroke distance (SD), and mean velocity (MV). All of the eight QFlow variables are shown ……Stroke distance x heartbeat / 60 (one cardiac cycle).” [Materials and Methods]
“Our MRI protocol performed QFlow scanning (perpendicular to blood flow and aortic curve)) to obtain two-dimensional images, which contained phase-shifting information. By drawing ROI on the vascular lumens (completely covering the true lumen and false lumen), it can obtain hemodynamic variables for statistical analysis. We set the flow direction to the head as positive flow. On the contrary, the flow direction to the foot was set as negative flow. Thus, our result revealed that the net blood volume (stroke volume, SV) in the aortic root and aortic arch was mainly contributed by the forward flow volume (FFV; toward the head direction). On the other hand, the net blood volume (stroke volume) in descending aorta and abdominal aorta was mainly contributed by the backward flow volume (BFV; toward the foot direction) (Figure 2A-C and Figure 4A-C ). This result is reasonable according to this study design and MRI protocol.” [Discussion]
[Comment 7] Did the authors detect any endoleak problems in their cohort with post-TEVAR MRI?
[Answer 7] We did find some endoleaks, as our supplement video shown.
[Comment 8] P1, Line 38-39: incomplete sentence (aortic disease imaging?)
[Answer 8] We have corrected the sentence.
“Presently the main imaging modality for detecting aortic diseases is computed tomography angiography (CTA).” [Introduction]
[Comment 9] Line 45: … are also a concern (?)
[Answer 9] We have corrected the sentence.
[Change]
“Further evaluation with contrast-enhanced cross-sectional imaging modalities, such as CT and MRI, is often used to evaluate aortic pathology. However, the major challenge is estimating the proper acquisition time for optimal contrast opacification of the target vessel.” [Introduction]
[Comment 10] There is no 1.2 part, so it is suggested to omit 1.1 title.
[Answer 10] We have omitted the 1.1 title as your comment.
[Comment 11] What is meant by back forward flow volume? Backward and forward flow? Identical patients sound strange.
[Answer 11]
The QFlow scanning is performed at four levels to obtain two-dimensional images (perpendicular to blood flow and aortic curve). By drawing ROI on the vascular lumens (completely covering the true lumen and false lumen), it can obtain eight hemodynamic variables for each ROI for the following statistical analysis. We set the flow direction to the head as positive flow. On the contrary, the flow direction to the foot was set as negative flow.
Stroke volume is the net volume of blood that passes through the contour of ROI during one cardiac cycle. Forward flow volume is the volume of blood that passes through the contour of ROI in the positive direction (toward the head direction) during one cardiac cycle. Backward flow volume is the volume of blood that passes through the contour of ROI in the negative direction (toward the foot direction) during one cardiac cycle.
We added Figure 1 and associated descript to clearly explain our study and MRI techniques.
“We performed QFlow analysis by drawing the region of interest (ROI) on the false lu-mens and true lumens at the following vascular segments: the aortic root, aortic arch, descending aorta, abdominal aorta at the level of the diaphragm, and abdominal aorta between the level of the celiac trunk and the superior mesenteric artery (SMA) (Figure 1). We set the flow direction to the head as positive flow. On the contrary, the flow di-rection to the foot was set as negative flow.” [Materials and Methods]
“Figure1. Illustration of QFlow scanning and drawing the region of interest (ROI). The QFlow scanning is performed at four levels to obtain two-dimensional images (perpendicular to blood flow and aortic curve). By drawing ROI on the vascular lumens (completely covering the true lumen and false lumen), it can obtain eight hemodynamic variables for each ROI for the following statistical analysis. We set the flow direction to the head as positive flow” [Figure 1]
“By drawing the ROI complete covering the vascular lumen, the computer could automatically generate analysis results of various variables. These variables include Stroke volume (SV), forward flow volume (FFV), backward flow volume (BFV), re-gur-gitant fraction(RF), absolute stroke volume (ASV), mean flux(MF, stroke distance (SD), and mean velocity (MV). All of the eight QFlow variables are shown ……Stroke distance x heartbeat / 60 (one cardiac cycle).” [Materials and Methods]

Reviewer 3 Report
This manuscript is a retrospective study of patients who presented with aortic dissection, and underwent 4D PC-MRI to evaluate aortic pathology. Overall, the manuscript is well written, but not clear about significance in data. Also some data are not clear how it was derived. Please clarify points stated below.
Major Questions:
- Figure 1-4:
All the graphs seem to have large standard deviation. Did you find significant differences? What are p-values? Please specify.
- Discussion:
The first paragraph usually summarize this study. This manuscript does not follow the regular order, therefore the Discussion is difficult to follow. Please reorganize the Discussion section.
Minor Questions:
- Page 1, line 26:
It is stated that “After TEVAR, the stroke distance increased in the TL, indicating more effective blood transport per heartbeat.” P-value is not presented. Was there a significant difference? Also why stroke distance is more important that total stroke volume (forward flow volume minus backward flow volume)?
- Page 5, Figure 1:
In descending and abdominal aorta, forward flow volumes are especially small with small standard deviations. Whereas Backward flow volumes are relatively large at TEVAR true lumen and AD true lumen. Why regurgitation fractions are very small?
- Page 6, Figure 2:
Please specify the abbreviation of AD.
- Page 6, Figure 2:
Stroke distance and mean velocities are negative which mean backward flow exceeds forward flow. But absolute stroke volume and mean flux are positive?
- Page 7, Figure 3:
In descending and abdominal aorta, forward flow volumes are especially small with small standard deviations. Whereas Backward flow volumes are relatively large especially in true lumen of Graft stent and bare stent. Why regurgitation fractions are very small?
- Figure 1-4:
The data on the very right are not labeled. Where did you measure? Please specify.
Author Response
Reviewer 3
This manuscript is a retrospective study of patients who presented with aortic dissection, and underwent 4D PC-MRI to evaluate aortic pathology. Overall, the manuscript is well written, but not clear about significance in data. Also some data are not clear how it was derived. Please clarify points stated below.
[Comment 1]
Figure 1-4:
All the graphs seem to have large standard deviation. Did you find significant differences? What are p-values? Please specify. [Answer 12] We revise this paragraph as your correction.
[Answer 1]
Response: We added Table 2 and Table 3 to demonstrate the detailed results of the QFlow measurements and redrawn the Figure2, Figure3, Figure 4, and Figure to clearly express the difference of the mean values between groups. Although there is a trend of deviation from the endovascular repair, most of the P-values are larger than 0.05 indicating no significant difference between groups. We added this weakness in the section of Study limitations.
[Change]
“First, The QFlow measurements presented a large standard deviation, and most of the P-values are larger than 0.05, indicating no significant difference between groups.” [Discussion]
[Comment 2]
The first paragraph usually summarize this study. This manuscript does not follow the regular order, therefore the Discussion is difficult to follow. Please reorganize the Discussion section.
[Answer 2]
We had adjusted the paragraph according to your suggestions
[Comment 3] Page 1, line 26:
It is stated that “After TEVAR, the stroke distance increased in the TL, indicating more effective blood transport per heartbeat.” P-value is not presented. Was there a significant difference? Also why stroke distance is more important that total stroke volume (forward flow volume minus backward flow volume)? [Answer 3] After TEVAR, the stroke distance increased in the TL (at descending and abdominal aorta, P=0.018 and 0.015), indicating more effective blood transport per heartbeat.
This study revealed that stroke distance is more effective than stroke volume to reflect the hemodynamic difference after TEVAR. We suppose this is because that stroke volume is more affected by the size of the vascular lumen. According to the algorithm, stroke distance is the net distance blood proceeds in the vessel during one cardiac cycle. Stroke volume is the net volume of blood that passes through the contour of ROI during one cardiac cycle. We noticed that the vascular lumens (included both false lumen and false lumen) were variable at a different vascular segment. This variability of luminal size at different vascular segments may affect the predictive power of stroke volume.
“After TEVAR, the stroke distance increased in the TL (at descending and abdominal aorta, P=0.018 and 0.015), indicating more effective blood transport per heartbeat.” [Abstract]
“This study revealed that stroke distance is more effective than stroke volume to reflect the hemodynamic difference after TEVAR. We suppose this is because that stroke volume is more affected by the size of the vascular lumen. According to the al-gorithm, stroke distance is the net distance blood proceeds in the vessel during one cardiac cycle. Stroke volume is the net volume of blood that passes through the con-tour of ROI during one cardiac cycle. We noticed that the vascular lumens (included both false lumen and false lumen) were variable at a different vascular segment. This variability of luminal size at different vascular segments may affect the predictive power of stroke volume.” [Discussion]
[Comment 4]
Page 5, Figure 1:
In descending and abdominal aorta, forward flow volumes are especially small with small standard deviations. Whereas Backward flow volumes are relatively large at TEVAR true lumen and AD true lumen. Why regurgitation fractions are very small?
[Answer 4]
QFlow analysis revealed that regurgitation fractions in the true lumens are consistently small in the true lumen. However, the backward flow volume is large, and the forward flow volume is small in the true lumens of the descending and abdominal aorta. The regurgitant fraction was automatically calculated as the fraction of the minor flow (usually the flow toward the heart) to the main flow (usually the flow away from the heart) that passes through the contour of ROI of the two-dimensional QFlow scanning. The backward flow (negative direction, toward foot) is the main flow characteristic of true lumens at the descending and abdominal aorta. Thus, the regurgitation fractions are still small.
The figure below is a demo case showing how to perform QFlow analysis. At ROI 2 (true lumen of aortic root), the regurgitation fraction was calculated as the fraction of backward flow volume to the forward flow volume(6.2/80.54=7.7%). At ROI 3 (true lumen of descending aorta), the regurgitation fraction was calculated as the fraction of forward flow volume to the backward flow volume(8.19/65.13=12.57%).
“Second, QFlow analysis revealed that regurgitation fractions in the true lumens are consistently small in the true lumen. However, the backward flow volume is large, and the forward flow volume is small in the true lumens of the descending and abdominal aorta. The regurgitant fraction was automatically calculated as the fraction of the mi-nor flow (usually the flow toward the heart) to the main flow (usually the flow away from the heart) that passes through the contour of ROI of the two-dimensional QFlow scanning. The backward flow (negative direction, toward foot) is the main flow char-acteristic of true lumens at the descending and abdominal aorta. Thus, the regurgita-tion fractions are still small.” [Discussion]
[Comment 5]
Page 6, Figure 2:
Please specify the abbreviation of AD.
[Answer 5] We added the abbreviation of AD.
[Comment 6]
Stroke distance and mean velocities are negative which mean backward flow exceeds forward flow. But absolute stroke volume and mean flux are positive?
[Answer 6]
Response: Stroke distance and mean velocity can be negative because that stroke distance and mean velocity reflect the "distance" (the flow direction to the head was set as positive flow) that blood proceeds in the vessels. On the contrary, absolute stroke volume and mean flux are positive because the absolute stroke volume was the absolute value of forwarding flow volume" plus the absolute value of backward flow volume, and mean flux reflects the stroke amount.
We have added detailed MRI methods and QFlow variable algorithms, as well as related discussions.
“We performed QFlow analysis by drawing the region of interest (ROI) on the false lumens and true lumens at the following vascular segments: the aortic root, aortic arch, descending aorta, abdominal aorta at the level of the diaphragm, and abdominal aorta between the level of the celiac trunk and the superior mesenteric artery (SMA) (Figure 1). …… Stroke distance x heartbeat / 60 (one cardiac cycle).” [Materials and Methods]
“Third, stroke distance and mean velocity can be negative because that stroke distance and mean velocity reflect the "distance" (the flow direction to the head was set as posi-tive flow) that blood proceeds in the vessels. On the contrary, absolute stroke volume and mean flux are positive because the absolute stroke volume was the absolute value of forwarding flow volume" plus the absolute value of backward flow volume, and mean flux reflects the stroke amount.” [Discussion]
[Comment 7] Page 7, Figure 3:
In descending and abdominal aorta, forward flow volumes are especially small with small standard deviations. Whereas Backward flow volumes are relatively large especially in true lumen of Graft stent and bare stent. Why regurgitation fractions are very small?
[Answer 7] This question is similar to the previous question (Page 5, Figure 1). We had answered this technique issue and added revision in the manuscript.
[Comment 8]
Figure 1-4:
The data on the very right are not labeled. Where did you measure? Please specify.
[Answer 8]
We revise the figures in this version , according to all reviewers’ comments.

Round 2
Reviewer 1 Report
no further comments
Author Response
Thanks for your encourage.
Reviewer 2 Report
This paper is a revision of the previously submitted paper on the 4D MRI measurements done pre- and post-TEVAR procedures. Authors have augmented the paper substantially, improved analysis and presentation of their data. However, there are still several suggestions/questions that are presented below:
- It is suggested to flip forward/backward flow definition to align with the natural flow patterns in the aorta. Specifically, flow from the heart to the legs to be set as forward, while from legs to the heart – as backward.
- I understand what is meant by the backward and forward flow, but still not sure what is meant by the “back forward flow” (Abstract: line 23-24, Results: line 207, Discussion: line 258).
- The paper would benefit from proofreading by a native English speaker.
Author Response
[Comment 1]
It is suggested to flip forward/backward flow definition to align with the natural flow patterns in the aorta. Specifically, flow from the heart to the legs to be set as forward, while from legs to the heart – as backward.
[Answer 1]
We set the flow direction from the heart to the legs as forwarding/positive flow. On the contrary, the flow direction from the legs to the heart was set as backward/negative flow. The QFlow data was analyzed, and the tables [Table 2-3] / figures [Figure 2-5] were remade. Figure legends were revised.
“We set the flow direction from the heart to the legs as forwarding/positive flow. On the contrary, the flow direction from the legs to the heart was set as backward/negative flow.” [Materials and Methods, line 120-122]
[Comment 2]
I understand what is meant by the backward and forward flow, but still not sure what is meant by the “back forward flow” (Abstract: line 23-24, Results: line 207, Discussion: line 258).
[Answer 2]
Response:The terminology "forward flow volume" was miswritten as "back forward flow" in these 3 sentences in the prior manuscript. We had corrected this error.
“TEVAR increased the forward flow volume of the true lumen (TL) (at abdominal aorta, P=0.047)." [Abstract, line 23-24]
“TEVAR increased the forward flow volume of the true lumen (TL).” [Results, line 207-208]
“TEVAR increased the forward flow volume of the true lumen (TL).” [Discussion, line 258-259).]
[Comment 3]
The paper would benefit from proofreading by a native English speaker
[Answer 3]
Thanks for your comment. This article has been reviewed by a native speaker this time and will be sent for English editing service later.

Reviewer 3 Report
I think the revised version reads better and presenting their results better. However there are some points need to be adjusted in order to avoid confusion.
Figure 2 and 4. In these two figures, is ‘Forward Flow’ a flow toward the head, and is ‘Backward Flow’ a flow toward the feet? If so, it is confusing, because ‘Forward flow’ in the aorta clinically means blood flow directing from the heart to distal. It is confusing when you define words that are commonly used in clinical setting in different way. Please fix it.
Figure 3 and 5. In general, forward flow means a flow in the direction from the heart to distal (e.g. feet), and it is positive. Please make an adjustment to avoid confusion.
Author Response
Reviewer 3
I think the revised version reads better and presenting their results better. However there are some points need to be adjusted in order to avoid confusion.
[Comment 1]
Figure 2 and 4. In these two figures, is ‘Forward Flow’ a flow toward the head, and is ‘Backward Flow’ a flow toward the feet? If so, it is confusing, because ‘Forward flow’ in the aorta clinically means blood flow directing from the heart to distal. It is confusing when you define words that are commonly used in clinical setting in different way. Please fix it.
Figure 3 and 5. In general, forward flow means a flow in the direction from the heart to distal (e.g. feet), and it is positive. Please make an adjustment to avoid confusion.
[Answer 1]
We set the flow direction from the heart to the legs as forwarding/positive flow. On the contrary, the flow direction from the legs to the heart was set as backward/negative flow. The QFlow data was analyzed, and the tables [Table 2-3] / figures [Figure 2-5] were remade. Figure legends were revised.
“We set the flow direction from the heart to the legs as forwarding/positive flow. On the contrary, the flow direction from the legs to the heart was set as backward/negative flow.” [Materials and Methods, line 120-122]
“(D) Regurgitant fraction (RF): The RF in the false lumen was higher in patients with graft stent than in those with bare stents.” [Figure 4, line 238-239]

Round 3
Reviewer 2 Report
The authors have adequately addressed my concerns.
Reviewer 3 Report
Thank you for your revision. I do not have any further questions.